

# Endogenic methylmercury in a eutrophic lake during the formation and decay of seston

Laura Balzer,[1*] Carluvy Baptista-Salazar,[2] Sofi Jonsson[2] and Harald Biester,[1]

[1] Institute for Geoecology, Environmental Geochemistry Group, Technische Universität Braunschweig, DE-38106
Braunschweig, Germany,

[2] Department of Environmental Science, Stockholm University, SE-106 91 Stockholm, Sweden

* *Corresponding author*: Laura Balzer (laura.balzer@tu-braunschweig.de)

**Abstract.** Anoxic microniches in sinking particles in lakes have been identified as important water phase production zones of monomethylmercury (MeHg) (endogenic MeHg). However, the production and decay of MeHg during organic matter (OM) decomposition in the water column and its relation to the total Hg concentration in seston are poorly understood. We investigated Hg speciation and chemical changes in sinking seston in a small and shallow (12-m-deep) eutrophic lake during phytoplankton blooms from April to November 2019. The results show that MeHg proportions are high in seston at the water
surface (up to 22 %) and at the oxic-suboxic redox boundary (up to 26 %). During suboxic OM decomposition, and with decreasing redox-potential, the concentration and proportion of MeHg in seston strongly decrease (< 0.5 %) as the water depth increases. Under these conditions, total Hg concentrations show a 3.8 to 26-fold increase. In the hypolimnion environment, changes in MeHg proportions were minimal in sinking seston, and samples collected by sediment traps had MeHg values similar to those measured at the sediment-water interface, though higher MeHg concentrations were found deeper in the
sediment. Our results indicate that cycling of MeHg and total Hg (THg) in seston within small productive lakes is largely controlled by the decomposition processes of settling seston and that the endogenic MeHg pool appears to be largely disconnected from the sedimentary MeHg pool.

## 1 Introduction

Lakes are dynamic systems where methylmercury (MeHg) is produced and can become bioaccumulated in the food chain
(Ravichandran, 2004). The methylation of inorganic divalent forms of Hg(Hg(II)) to toxic MeHg is carried out by a variety of obligate anaerobic microorganisms (Peterson et al., 2020; Gilmour et al., 1992; Fleming et al., 2006; Gilmour et al., 2013) that decompose the organic matter (OM) produced mainly by the primary productivity occurring in the water column. Thus, lacustrine Hg methylation is, presumably, closely connected to the biological pump and related changes in redox conditions, as well as being influenced by temporal and spatial variabilities. Aquatic MeHg production has been previously investigated
in sediments (Jensen and Jernelöv, 1969; Gilmour et al., 1992; Robinson and Tuovinen, 1984; Hammerschmidt et al., 2004;



Hollweg et al., 2009; Sunderland et al., 2004; Bouchet et al., 2013) and more recently in the oxic and anoxic water column (endogenic source) of marine (Topping and Davies, 1981; Mason and Fitzgerald, 1990; Heimbürger et al., 2010; Soerensen et al., 2018; Sunderland et al., 2009; Cossa et al., 2011; Cossa et al., 1997; Heimbürger et al., 2015; Schartup et al., 2015a; Munson et al., 2018; Monperrus et al., 2007; Bouchet et al., 2013; Ortiz et al., 2015; Kirk et al., 2008; Hammerschmidt and

Bowman, 2012; Blum et al., 2013; Cossa et al., 1994; Malcolm et al., 2010; Wang et al., 2018) and freshwater systems (Gascón Díez et al., 2016; Lepak et al., 2018; Eckley et al., 2005; Watras et al., 1995; Eckley and Hintelmann, 2006; Peterson et al., 2020; Mauro et al., 2002; Henry et al., 1995; Gascón Díez et al., 2018; Watras and Bloom, 1994). Research conducted in the 1980s in marine systems (Topping and Davies, 1981) and in lake systems in the 1990s (Watras and Bloom, 1994) suggested that the endogenous MeHg sources are similar to, or may even exceed, sedimentary MeHg production. Endogenously produced

MeHg may increase MeHg exposure at the base of the aquatic food chain which may explain the high MeHg levels observed at higher trophic levels (Heimbürger et al., 2015; Gallorini and Loizeau, 2021). Several studies indicate that maximum MeHg concentrations occur at the middle of the water column. The observed changes in water column MeHg concentrations were related to OM mineralization (Sunderland et al., 2009; Cossa et al., 2011; Cossa et al., 1997; Heimbürger et al., 2010; Heimbürger et al., 2015) below the oxic/anoxic boundary(Mauro et al., 2002) or along redox gradients in the water

column(Peterson et al., 2020; Watras et al., 1995). Based on observations of water column MeHg formation under differing states of oxygen ($O_2$) saturation, anoxic microniches have been proposed as important MeHg production sites in and around settling particles in both lacustrine and the marine water columns (Gascón Díez et al., 2016; Schartup et al., 2015a). However, MeHg formation in anoxic microniches has only been observed directly in marine particles (Ortiz et al., 2015; Gallorini and Loizeau, 2021). Regarding the abundance, chemical composition and microbial colonization, aggregates of settling particles

formed in lakes ("lake snow") do not differ significantly from those in marine waters ("marine snow") (Grossart and Simon, 1993). This implies that anoxic microniches can also be formed in lakes, particularly in eutrophic lakes with high OM turnover. Previous studies on Hg cycling in mesotrophic and eutrophic lakes have shown that high primary productivity enhances lacustrine Hg sedimentation due to algae scavenging water column Hg (Schütze et al., 2021; Biester et al., 2018). Hg scavenged by aquatic microorganisms and particles may undergo a process of species transformation concurrent with the OM degradation

processes that occur during particle settling (Meyers and Eadie, 1993).

Degradation is most pronounced in recently produced OM (Higginson, 2009), which is primarily altered in the upper water column (Meyers and Eadie, 1993). Meyers and Eadie(Meyers and Eadie, 1993) showed that OM at different water column depths is decomposed by different biogeochemical processes. Degradation rates decrease with depth, resulting in increasing amounts of refractory OM (Higginson, 2009). The temporal and spatial occurrences of lacustrine MeHg in settling particles

and how it changes during OM decomposition throughout the water column are still poorly understood. Most studies are either based on laboratory analysis (Pickhardt and Fisher, 2007), studies using sediment traps (Gascón Díez et al., 2016) or analysis





of bottom sediment(Zaferani et al., 2018). Moreover, nearly all of these studies are based on singular sampling from only a few locations in the water column. These low resolution sampling schemes limit our ability to decipher the temporal and depth dependent processes of MeHg formation or degradation in sinking sestons. In addition, most existing studies were carried out

in oligotrophic boreal lakes. This is important as many lakes outside the boreal zone have high nutrient levels or suffer from eutrophication and often contain comparatively high Hg concentration levels in the sediment (Schütze et al., 2021).

In this study, we developed water column and seston depth profiles (at 1 m intervals) in a productive lake over seven days between April and November (2019). This was done to investigate the formation and fate of MeHg during algal blooms and during the degradation of settling seston throughout seasonal lake cycle. We hypothesize that the high primary productivity in

eutrophic lakes and related changes in water column redox conditions provide suitable condition for MeHg scavenging or MeHg formation by or in seston. Moreover, we expect that the MeHg concentrations and proportions in seston are altered through OM degradation in the water column and that the endogenic MeHg pool is likely independent from sedimentary MeHg formation.

## 2 Materials and Methods

### 2.1 Study site and sampling

Lake Ölper (105 m a.s.l., 52°28′80″ N, 10°51′20″ E) is an artificial dimictic lake located in the city of Braunschweig, Germany. We used this lake as a natural laboratory, as it has no direct inflow and receives relatively low surface run-off, which minimizes the external input of MeHg. Moreover, the lake is known to show negligible lateral currents, which may cause frequent mixing. The lake has an area of 0.158 km² and is 10–13 m deep. It is mainly fed by rain and groundwater and receives input from the

nearby Oker River during rare flood events only via a small trench (Galggraben). There were no flood events in 2019 and surface run-off was minimal due to very arid conditions. The lake is surrounded by a park, and its catchment is flat and is smaller in area than the lake itself. The vegetation is dominated by willows. Ground elder and reeds are distributed heterogeneously around the lake (mostly present on the south bank). Water and seston samples were taken on seven days between April and November 2019 using a clean stainless steel immersion pump (Comet Combi 12–4T). The water column

was sampled over the deepest (~12 m) portion of the lake. Samples were collected from the surface down to the sediment-water interface at 1 m intervals. Samples for chlorophyll a determination, following DIN 38409 - H 60 (Deutsches Institut für Normung, 2015), were stored in 0.5 L and 2 L PE flasks that had been prerinsed and cleaned with acid. Samples intended for chemical analyses (i.e., THg, DOC, Mn, Fe, major ions) were vacuum filtered using <0.45 µm nylon filters and stored in 50 ml PE Falcon tubes for.



Seston samples were collected by means of a plankton net with a 25-µm mesh and placed into 100-ml acid-cleaned and prerinsed flasks. Surface seston samples were collected at 0.2 m depth by drawing the net behind the boat. Seston samples from deeper water layers (up to 4–12 m) were collected by pumping between 60 L and 120 L through the plankton net. Depending on the plankton abundance and distribution in the water column, sampled depths varied by day between 4 and 12 m, but in most cases covered the upper 4 m (except in November where only the upper 2 m were sampled). Additionally, the

change in redox conditions between well oxygenated and poorly oxygenated water also impacted the sampling depth. All seston samples were frozen immediately after sampling and subsequently freeze-dried and homogenized with a glass pestle for further analyses (THg, MeHg, CNS). Water temperature, electrical conductivity (EC), pH and $O_2$ concentrations and saturation were measured in situ using handheld single-parameter probes.

Bulk settling seston was collected over 141 days from early May to late September (06.05–24.09) by means of a sediment trap

positioned at the deepest part of the lake approximately 1 m above the sediment-water interface (~ 11 m depth). The sediment trap consisted of two individual particle interceptor traps (PIT). Each trap had a collection area of 0.45 cm² and was kept vertical in the water column by two buoys. Prior to deployment, each PIT was acid-cleaned and prerinsed with lake water. In addition, a short sediment core (6 cm) was taken using a UWITEC gravity corer at the sediment trap position. Sampling material was frozen immediately after collection, freeze-dried and ground for further analyses (THg, MeHg, CNS).

## 2.2 Analyses of water samples

### 2.2.1 Total dissolved Hg

For total dissolved Hg analyses, filtered water samples were stabilized using a 0.5 % (v/v) ultrapure BrCl–HCl solution in order to convert all organic Hg compounds to inorganic $Hg^{2+}$. THg was determined by means of cold vapour atomic fluorescence spectrometry (CV-AFS, Mercur Analytic Jena AG, Germany) after $Hg^{2+}$ was reduced with stannous chloride

according to EPA method 1631. The quality of the measurements was controlled by the certified reference Material ORMS 5 "elevated mercury in river water" (NRC CNRC) with a Hg concentration of 26.2±1.3 ng $L^{-1}$. The mean recovery was 102.5 % (n= 20).

### 2.2.2 Iron and manganese

Dissolved Fe and Mn concentrations were analysed in the acidified water samples (1 % (v/v) bidistilled nitric acid ($HNO_{3;}$))

by means of inductively coupled plasma–optical emission spectrometry (ICP–OES, Varian 715 ES, Agilent Technologies Inc., USA).

The quality of the measurements was controlled by CRM SLRS-6 (river water) with an Fe concentration of 84.3±3.6 µg $L^{-1}$ and a Mn concentration of 2.12±0.1 µg $L^{-1}$. The mean recovery of the CRM (n= 6) was 94.4 % for Fe and 112.1 % for Mn.





### 2.2.3 Dissolved organic carbon

DOC concentrations were determined after water samples were acidified with HCl to a pH value of two to remove carbonic acid and after thermocatalytic oxidation of the sample by means of a TOC-Analyser (multi N/C 2100, Analytic Jena AG, Germany). Measurements of DOC were validated using CRMs (ION-96.4, TOC20 and NW-Ontario-12), with mean recoveries of 107.7 % (n= 11), 95.3 % (n= 17) and 97 % (n= 9).

### 2.2.4 Nitrate and sulphate

$NO_3^-$ and $SO_4^{2-}$ were determined by means of ion exchange chromatography (761 Compact IC, Metrohm AG, Switzerland). The quality of the measurements was controlled by the CRM, Roth Multi-Element Standard solution ($NO_3^- = 24.997 \pm 0.064$ mg $L^{-1}$; $SO_4^{2-} = 30.049 \pm 0.070$ mg $L^{-1}$) and the IC Multi-Standard solution 02179 (Bernd Kraft) ($NO_3^- = 1000$ mg $L^{-1}$; $SO_4^{2-} = 1000$ mg $L^{-1}$). Mean recoveries for the CRM Roth (n =20) were 97 % for $NO_3^-$ and 96.7 % for $SO_4^{2-}$ and for the CRM IC Multi-Standard solution 02179 (Bernd Kraft) (n=3) 86 % for $NO_3^-$ and 91.1 % for $SO_4^{2-}$.

### 2.2.5 Chlorophyll a.

Chlorophyll was determined according to the German standard procedure DIN 38409 – H 60 (Deutsches Institut für Normung, 2015). Between 1 and 2 litres of lake water were vacuum filtered using Whatman GF/F filters (Carl Roth GmbH + Co. KG, Germany). The filters were folded and homogenized, and the pigments were extracted immediately with 15 ml 90% ethanol (78 °C) in a opaque 50-ml Falcon tubes and stored for 12-24 h in the dark at room temperature. These extracted pigments were

clarified by filtration using membrane filters, and their concentration was measured by means of a UV–VIS spectrometer Lambda 25 (Perkin Elmer) at 750 nm and 665 nm against 90% ethanol. The concentration was corrected for phaeopigment by acidification of the sample with 0.3 Vol% 2 M HCl (Deutsches Institut für Normung, 2015).

### 2.3 Analyses of seston, sediment trap material and bottom sediments

Seston samples, sediment traps and core materials were freeze dried (LYOVAC GT 2- E). The dried seston samples were

homogenized using a glass stick. Sediment trap and core samples were ground in a cleaned agate ball mill. All the solid samples were subjected to the same methods for the following analyses.

### 2.3.1 Total Hg

The THg content in all solid samples was analysed by thermal decomposition followed by preconcentration of Hg on a gold trap and atomic absorption spectrometry using a DMA-80 direct mercury analyser (Milestone, Italy). For quality control, three



standard reference materials with different matrices (apple leaves NIST-1515 (THg= 44 ng g$^{-1}$), Chinese sediment NCS DC 73312 (THg= 40 ng g$^{-1}$) and plankton material BCR-414 (THg= 276 µg g$^{-1}$) were measured.

The mean recovery of the CRMs was 111.9 % for apple leaves (n= 20), 93.6 % for Chinese sediment (n= 24) and 107.7 % for plankton material (n= 9).

### 2.3.2 Methylmercury

The extraction of MeHg from the seston samples was performed using a slightly altered procedure for biota samples suggested by the U.S. Geological Survey's Mercury Research Laboratory (USGS Method 5 A-7) (USGS-Mercury Research Laboratory, 2016). Samples were digested in 5 M nitric acid (instead of 4.5 M) for ~15 h (instead of 8 h) until no visual residues could be observed to ensure complete digestion. Digested samples were buffered with sodium acetate at pH ~4.9 and ethylated using sodium tetraethylborate (NaTEB). MeHg was analysed using a purge and trap CV-AFS (Tekran 2700) methylmercury

analyser.

The quality of the measurements was controlled by three CRMs, TORT-2 lobster hepatopancreas (MeHg=163.4 ng g$^{-1}$), DOLT-5 dogfish liver (MeHg=127.9 ng g$^{-1}$) and SRM® 15566b oyster tissue (MeHg=14.2 ng g$^{-1}$). The mean recoveries for the CRMs were 83.11 % for TORT-2 (n= 7), 108.33 % for DOLT-5 (n=6), and 103.6 % for SRM® 15566b-2 (n= 7).

To extract MeHg from the sediment samples, between ~0.5 – 1 g of material was weighed into new 50 mL Falcon tubes and

between 20 – 100 µL of an internal standard, an isotopically enriched Me$^{200}$Hg standard with concentration 1.1 ng g$^{-1}$, was added and then left to equilibrate for an hour. After equilibration, 10 mL KBr (1.4 M), 2 mL CuSO$_4$ (2 M) and 10 mL dichloromethane, DCM (CH2Cl2) were added to each tube, which was the capped and left for 45 min. To extract MeHg, the samples were rotated at 85 RPM on a sample rotor for 45 min and then centrifuged for 5 min at 3000 RPM. MeHg was analysed using a Tekran® Model 2700 Automated Methylmercury Analysis System connected to an Inductively Coupled Plasma Mass

Spectrometer, Thermo–Fisher X- series 2 (ICPMS). Prior to analysis, half the extracted sample was ethylated using sodium tetraethylborate (NaTEB) at pH 4.9 (using 225 µl of a 2 M acetate buffer). The certified reference material (ERM-CC580, estuarine sediment) analysed was on average 110 % of the certified value (75 ± 4 ng g$^{-1}$) (for a detailed description, please see the method section in the supplementary material).

### 2.3.3 Carbon, Nitrogen, Sulphur

Total C, N and S in all solid samples were measured by means of an elemental analyser (EuroEA 3000, Hekatech GmbH, Germany) that combusted 10–20 mg aliquots of each sample in a tin capsule calibrated with a sulphanilamide standard (C= 41.75 ± 0.17 %; N= 16.26 ± 0.22 %; S= 18.64 ± 0.18 %) and BBOT (2.5-Bis (5-tert-butyl-benzoxazol-2-yl)thiophene); C= 72.52 %; N= 6.51 %; S= 7.44 %). The quality of the measurements was controlled by three CRM, NIST 1515 apple leaves



(N = 2.25 ± 0.19 %), NCG DC 73030 Chinese soil (C= 0.617 ± 0.044 %; S= 0.2 ± 0.03 %), MOC soil standard (C= 3.19 ±

0.07 %; N= 0.27 ± 0.02 %; S= 0.043 ± 0.005 %) and sulfanilamide (1 for every 10 analyses).

Mean recoveries for the CRMs in sulfanilamide (n= 15) were 99.7 % for C, 98.8 % for N and 106.4 % for S; NIST 1515 apple

leaves (n= 4), 95.4 % for N; NCG DC 73030 Chinese soil (n= 2), 97.9 % for C and 108.3 % for S; MOC soil standard (n= 4),

80 % for N; 99.9 % for C and 106.4 % for S.

## 3 Results and Discussion

### 3.1 Changes in phytoplankton productivity and redox conditions

Based on the chlorophyll a (Chl) concentration, pH and $O_2$ concentration results, we found two productive periods. The month

of April and the months from June to September were defined as periods of high phytoplankton productivity, with chlorophyll

a concentrations between 29 µg $L^{-1}$ in June and 95 µg $L^{-1}$ in September (Fig. S2; August 12 Chl. n.d.), supersaturation of $O_2$

with values between 115 % and 151 % (except at the 2 m depth in June where the value was 66 %) and high pH values between

8.7 and 9.3 (except at the 2 m depth in June which was 7.8). The highest pH values occurred in the upper two metres of the

water column (0-2 m) on August 12 and 19 and were 9 and 9.3 respectively (Fig. 1). May and November are defined as the

low phytoplankton productivity period. In May, the surface pH dropped from 9 to approximately 7.7, while $O_2$ saturation

dropped to between 80 and 86.6%. Chlorophyll a concentrations were 2.5 to 2.8 8 µg $L^{-1}$, indicative of low phytoplankton

productivity. The surface water pH dropped to 7.3 after mixing in November, when productivity was very low (Fig. 1) (Chl

n.d.), with correspondingly low $O_2$ saturation values throughout the entire water column (mean: 28.8 %).

The decrease in pH and $O_2$ saturation with depth indicates the zone where OM respiration and other oxidative processes began

to exceed primary production and where the onset of related changes in redox conditions occur. Bacteria use a fixed sequence

of alternative electron acceptors for OM mineralization ($O_2$ > nitrate ($NO_3^-$) ~ $Mn_{ox}$ > $Fe_{ox}$> sulphate ($SO_4^{2-}$)) (Froelich et al.,

1979). Thus, the progressive decay of sinking OM is indicated by the concentration profiles of $O_2$, dissolved $NO_3^-$, dissolved

Mn and Fe. Lake Ölper shows a clinograde seasonal depth profile of dissolved $O_2$ from April to September, with a sharp

oxycline that started to develop in May at a depth of approximately 4 m (Fig. 1). The oxycline occurs at variable depths within

the first 4 metres until September when there is a decrease in $O_2$ saturation to less than 10 % within 1–3 m. There is an increase

in Mn concentrations in the zone where the sharp drop in $O_2$ saturation occurs. This section of changing electron acceptors is

hereafter defined as the redox transition zone (RTZ) from oxic to suboxic conditions. The position (1 - 4.5 m) and thickness

(1 – 3 m) of the RTZ changed throughout the sampling period and were controlled by the intensity of productivity in the

surface layer and its progressive decay during sinking. Mn reduction was first detected in April and then increased and

ascended in the water column from May to September, whereas Fe reduction was only detectable below ~ 8 m in May and





below 9 m in August and September. After mixing in November, the Mn and Fe concentrations were uniformly low (Fig. 1). This indicates that during the high phytoplankton productivity period, primary production and OM decay facilitated Mn and, to a lesser extent, Fe reduction (Fig. 1), whereas $SO_4^{2-}$ reduction was not observed (Fig. S3).

## 3.2 Microbial decomposition of OM in the water column as indicated by C/N ratios

Carbon to nitrogen ratios (C/N) in lake sediments or seston are commonly used to distinguish between autochthonous and allochthonous-derived OM. Seston C/N ratio in Lake Ölper showed a median of 6.4 (n= 42; 4–8.3; excluding two outliers with values of 12.6 and 16.3 in bulk sampled seston in August) (Fig. 2), indicating plankton as the dominant OM source (Müller, 1977).

However, for water column seston samples at individual days, relative changes in C/N ratios in seston can be used as an indicator for OM degradation (Meyers and Lallier-Vergés, 1999). In the photic zone, C/N ratios decreased slightly with depth, indicating selective C mineralization under oxic conditions and relative enrichment of N. Under suboxic conditions, C/N ratios were lowest within the RTZ before they strongly increased again below the RTZ (Fig. 2) due to preferential decomposition of nitrogen-rich proteins resulting in a relative enrichment in C (Meyers and Lallier-Vergés, 1999) (though this was not observed in September). Here, the C/N ratio showed the greatest change, so we suggest that microbial decomposition and the alteration of OM were more intense within and just below the RTZ compared to the layers above. It is generally accepted that the C/N ratio of planktonic OM increases steadily while decomposing and sinking down to the sediment (Gordon, 1971; Müller, 1977). Here, the decomposition of seston during vertical transport in the water column must be divided into zones of aerobic and anaerobic decay, as suggested by Oguz et al. (2000), with the transition zone (RTZ) in between. Despite the relatively higher loss of N, both C and N strongly decreased within and below the RTZ in June, August 12 and 19, and September, while sulphur (S) increased (Fig. 2). It was noted that samples collected closer to the sediment-water interface in June, August 12 and 19, and September, had C and N concentrations and C/N ratios that were similar to those observed in the sediment trap material (Fig. 1) and the upper sedimentary layer (0-2 cm: C: 8.8%; N: 0.7 %; S: 0.8 %; C/N: 12.76). Thus, we assume that decomposition appeared to decrease further below the RTZ as OM became more refractory with during decomposition. Similar to Lake Ölper, a maximum of OM decomposition beneath the bottom of the euphotic zone was reported by Saino and Hattori (1987). Such a thin layer with enhanced microbial activity is likely caused by changing redox acceptors and the aggregation of labile OM enriched by nutrients accumulating in areas with neutral buoyancy, which was likely induced by water column stratification during phytoplankton blooms(Schartup et al., 2015a). In April, the increase in decomposition-related C/N ratios occurred continuously below 4 m when $O_2$ saturation was still relatively high (see also the change in colour of the seston from a light green to darker green–brownish colour at ~6 m (Fig. S6)). Here, no decomposition maximum, as indicated by a sharp change in the C/N ratio, could be observed because the neutral buoyancy induced by water column stratification was not





sufficiently developed. The C/N ratios in seston indicate that degradation of OM generally occurs quickly in the upper aerobic water layers, including the RTZ, and probably slower below the RTZ. This may indicate that the influence of OM

decomposition on MeHg and THg concentrations is strongest in the oxygenated upper waters, especially in stratified water columns.

## 3.3 Spatial and temporal distributions of MeHg and THg

The seston MeHg concentration in the upper water layer (upper two metres) was between 1.6 ng g$^{-1}$ in April and 48 ng g$^{-1}$ in November (corresponding to 0.5 %–18.6 % MeHg of THg (MeHg-%)) (Fig. 3). April, June, August and September, when

primary production was high, showed comparatively low MeHg concentrations (1.6–11.4 ng g$^{-1}$) in the upper two metres. Higher MeHg concentrations were found in May and November (15.6–48 ng g$^{-1}$) when lower pH (<7.7) indicated low primary productivity (Fig. 1, 3 and 4). Seston samples from the period of low phytoplankton productivity (May and November) might be dominated by zooplankton, as the seston from the upper water layers showed a comparably higher N content and thus lower C/N ratios (N: 7.5–11 %, C/N: 4.0–4.5) than seston from April, June, August and September (N: 2.5–6.5 %, C/N: 5.2–8.3)

(Fig. 4) (compare Fig. S6). This would explain the comparatively elevated MeHg in periods of low phytoplankton productivity, as biomagnification from phytoplankton to zooplankton is known to increase MeHg concentrations(Wang et al., 2018).

In April, when no RTZ was established, MeHg concentrations showed a continuous increase with depth as O$_2$ concentrations decreased and had a comparatively low maximum of 7.6 ng g$^{-1}$ at 9 m depth (Fig. 3).

During periods in which the RTZ was clearly defined, MeHg concentrations in seston showed a pronounced maximum at the RTZ that did not occur in April, when no RTZ was observed (Fig. 3). Thus, from May to August, MeHg concentrations showed a continuous increase from the surface to the RTZ, where the values peaked (Fig. 3). Below the RTZ, MeHg concentrations decreased 1.3 to 4.4-fold even though the water column became progressively anoxic. In September, MeHg concentrations peaked just below the RTZ. However, not enough material for seston analyses could be obtained from deeper layers.

The high MeHg concentrations at the RTZ could be explained by settling seston that aggregates within the RTZ (neutral buoyancy induced by water column stratification)(Schartup et al., 2015a). Settling particles such as this may form anoxic microniches, providing a thin vertical layer of high Hg methylation and biological activity, as suggested in other studies (Gascón Díez et al., 2016; Schartup et al., 2015a; Ortiz et al., 2015). In April, when no RTZ and no stratification-induced neutral buoyancy were observed, the formation of aggregates and thus the development of anoxic MeHg microniches was

unlikely. This would explain the continuous increase in MeHg concentrations with depth as opposed to the pronounced mid water column-maximum observed when redox conditions became increasingly anoxic with water depth (compare the dissolved



O$_2$, Fe and Mn concentrations in Fig. 1). There might be other explanations for the MeHg midwater-maximum that we cannot completely exclude, but we assume those to be less critical than the formation of MeHg in anoxic microniches.

First, MeHg is transported from other sources, such as bottom sediments or the littoral zone, into the RTZ. This is likely to be
negligible, as the depth profiles of O$_2$, Fe, Mn, pH and EC indicate that mixing is minor during times of high productivity. Second, Hammerschmidt and Bowman (2012) suggested that there might be other methylators not yet identified that are able to methylate Hg even under oxic or suboxic conditions. Third, the MeHg concentration maxima observed in the RTZ may lie within the habitat of herbivorous and predatory zooplankton that graze the algal biomass. Higher amounts of zooplankton would increase the MeHg concentration in our seston sample due to biomagnification from phytoplankton to zooplankton.
However, the increase in N concentrations with depth was relatively small (between a 1.1 and 1.4-fold increase from the surface to the highest N concentration in the RTZ) compared to changes in N concentration between individual sampling days (up to a 4.4-fold increase at the surface layer) (Fig. 2). Thus, zooplankton occurrence appears to increase MeHg concentrations between individual sampling days, but the effect within individual depth profiles is likely minor and impact the MeHg maxima in the RTZ only marginal. Thus, changes in N and C concentrations with depth are predominantly a result of OM
decomposition.

The reasons for the high proportion of MeHg in the seston of the surface layer above the RTZ (up to 22 %) when O$_2$ saturation was high is unknown. At high O$_2$ concentrations in the surface layers, the flux of O$_2$ into settling particles is assumed to be higher than the consumption of O$_2$ inside the particle. Thus, O$_2$ concentrations at the surface layer are too high to form anoxic microniches. One possible explanation is that due to vertical mixing above the RTZ, MeHg produced in the RTZ may be
transported to the surface layer where it is taken up or adsorbed by phytoplankton (Kirk et al., 2008). This would explain the elevated MeHg-% even under supersaturated O$_2$ concentrations in the surface layer.

We suggest that the MeHg midwater maximum in Lake Ölper results from enhanced microbial activity at strong redox gradients related to high biomass production in the surface layers and intense decomposition at layers with neutral buoyancy. The midwater maxima near the surface and the euphotic zone are likely to enhance MeHg exposure to the lacustrine food web.

**3.4 Changes in THg/MeHg ratios in seston along redox gradients**

The median THg concentrations in the seston were 0.2 µg g$^{-1}$ (0.03–1.2 µg g$^{-1}$), with 0.13 µg g$^{-1}$ above the RTZ and 0.36 µg g$^{-1}$ within and below the RTZ (Fig. 3). In five out of seven sampling days (excluding May and November), THg concentrations clearly increased from the surface to the deepest sampling point by factors of 3.8–26.4, with the highest increase occurring within or especially below the RTZ. The greatest enrichment was observed in September, when THg concentrations were
specifically low in seston from the surface layer. Due to the concentrations of dissolved Hg in the water column of Lake Ölper being relatively constant and low over time and throughout the lake, we conclude that water phase Hg is unlikely to be the





source of Hg enrichment in sinking seston. Concomitant with the THg enrichment, C was depleted 1.2 to 3.9-fold and N 1.2 to 7.7-fold. This loss of mass (loss of N and C) during OM decomposition indicates that THg is somewhat relatively enriched in seston as a result of mass loss.

As discussed in the previous section, the MeHg concentration showed a midwater maximum resulting from enhanced methylation at strong redox gradients (Fig. 3). We hypothesize that above the midwater-maximum, MeHg progressively accumulates in sinking seston from the water phase until it reaches the redox boundary, where it accumulates in areas where neutral buoyancy is induced by water column stratification. The decreasing MeHg concentration below the midwater-maximum cannot be explained by decreasing MeHg production alone. Otherwise, we would expect a relative enrichment of
MeHg in seston due to carbon loss, as observed for THg.

The varying MeHg concentrations in seston can also not be explained by changing dissolved organic carbon (DOC) concentrations ($r= 0.03$; $p= 0.84$). However, recent studies have shown that fresh algae-derived dissolved organic matter (DOM) can enhance MeHg uptake (Schartup et al., 2015b). Due to the absence of continuous inflow, DOM in Lake Ölper is mainly produced in situ by seston decomposition since the catchment influx is low. DOM may play a crucial role in MeHg
uptake in oligotrophic lakes or lakes with high catchment runoff but does not appear to play a crucial role in Lake Ölper. Hence, there must have been a loss of MeHg from the seston within and below the RTZ, which reduces MeHg in the sinking seston. Although we did not determine demethylation rates directly, we conclude that OM decomposition in seston is accompanied by intense MeHg demethylation during which Hg is released into the water or remains and is reabsorbed by seston in a different Hg form. The increase in THg concentrations in the seston within and below the RTZ supports the latter
hypothesis.

It is unknown how Hg is bound in degraded OM. It has been shown that microbial $SO_4^{2-}$ reduction, which produces sulphide, occurs in settling particles within oxygen-deficient zones and in microenvironments within the centre of suspended particles (Raven et al., 2021; Shanks and Reeder, 1993). The sulphide produced may form insoluble complexes with Hg (Shanks and Reeder, 1993; Bianchi et al., 2018), such as Hg sulphides (HgS), meaning that Hg becomes less available for methylation
(Zhang et al., 2012). As S concentrations in the sinking seston of Lake Ölper strongly increased, within and especially below the RTZ (Fig. 2), the formation of Hg sulphides appears likely.

Suboxic decomposition processes of sinking OM and related redox conditions change THg to MeHg in sestons within a few metres during summer stratification. Thus, THg and MeHg fluxes to the sediment are largely determined by changes in OM composition and mass loss during decomposition.



## 3.5 THg and MeHg in sediment traps and the upper bottom sediment layers

Seston that settled in the sediment trap contained 1.5 µg g$^{-1}$ THg and 7.8 ng g$^{-1}$ (0.5 %) MeHg. The undecomposed seston from the upper two metres exceeded the MeHg concentration in the sediment trap material by a factor of 1.2 (median) and by up to 3-fold in the RTZ. The uppermost layer of lake-bottom sediment (0–2 cm) shows THg (2.2 µg g$^{-1}$) and MeHg (5.86 ng g$^{-1}$) concentrations comparable to those of the sediment trap material. However, with values of 39 ng g$^{-1}$ (2–4 cm) and 26 ng g$^{-1}$ (4–6 cm), the deeper sediment layers show higher MeHg concentrations at similar THg concentrations (2.06 and 1.96 µg g$^{-1}$) than the trap material, which indicates that in deeper sediment layers, there might be Hg methylation independent of the endogenic Hg methylation and/or fluxes to the water that are lower than methylation rates.

On the other hand, THg was up to 41 times higher in the sediment trap material than in seston recovered from the upper two metres (0.04–0.4 µg g$^{-1}$) and between 1.3 and 20 times higher than in the seston (0.08–1.2 µg g$^{-1}$) sampled below the first two metres. The MeHg and THg concentrations and proportions in the deepest seston samples indicate that the closer to the sediment the seston was collected, the closer the chemical composition (C, N concentrations) and the THg/MeHg ratio were to that of the surface sediments. In-trap mineralization may further increase the relative enrichment of THg and loss of MeHg compared to seston at the water surface. Assuming a linear regression of OM fraction loss of -0.001864 per day, as calculated by Radbourne and Ryves (2020), the initial OM content in our sediment trap material may be reduced by 26.3 % after 141 days. This highlights the importance of water column OM decomposition on MeHg cycling in lakes and the need for seston sampling at high spatial and temporal resolutions since sediment traps do not represent all the potential variations in Hg and MeHg cycling in lakes. Hg scavenging and OM decomposition control which Hg species are transported from surface euphotic waters into deeper water and sediments, and the extent to which this occurs. Due to the absence of water mixing during summer stagnation, the decreasing MeHg concentration in seston below the RTZ and the comparable low MeHg concentration in the sediment trap material, MeHg derived from sediments is unlikely to be the source for the MeHg in seston. This is further supported by the lower MeHg concentrations in the uppermost sedimentary layer compared to those in deeper in the sediment. Thus, in this small eutrophic lake, the epilimnetic pathway is likely of greater importance for MeHg bioaccumulation in the trophic web than the MeHg benthic-hypolimnetic pathway. Additionally, both pools appear to be largely disconnected, as suggested in a recent review by Gallorini and Loizeau (Gallorini and Loizeau, 2021).

## 4 Conclusions

We showed that separation between the sediment- and water-phase MeHg sources occurred not only in deep lakes but also in relatively shallow (12–13-m-deep) and productive lakes. Here, internal lake productivity and summer stratification strongly control MeHg production within the predominantly oxic and suboxic water column.





We propose that the intensity of phytoplankton productivity, OM decomposition along redox gradients and the formation of
summer stratification and anoxic microniches in sestons are more crucial than depth in terms of MeHg production and Hg
scavenging in eutrophic lakes. We further conclude that the decomposition of labile OM and the demethylation or loss of
MeHg within the RTZ exert a stronger influence on MeHg fluxes into the sediment than do anoxic conditions below the RTZ.
Water column MeHg formation and degradation in eutrophic lakes appears to be intense and occurs rapidly and at rates similar
to what we observed within the bottom sediments. The predicted increase in lake water temperatures and nutrient fluxes will
enhance phytoplankton productivity and eutrophication in many lakes, which in turn will lead to declining $O_2$ saturation (Ho
et al., 2019; Jane et al., 2021). According to our findings, such changes likely facilitate MeHg bioaccumulation in the trophic
web due to increased endogenic MeHg production, especially in small eutrophic lakes.

**Data availability:**
The authors declare that all data supporting the findings of this study are available within the paper and its supplementary
information files.

**Author contributions:**
L.B. and H.B. designed the study. L.B., C.B.S. and S.J. carried out the laboratory work. L.B. carried out all sampling and data
analyses. L.B. and H.B. wrote the manuscript, and all authors edited and revised the manuscript.
**Competing interests:** The authors declare that they have no conflicts of interest.
**Correspondence** and requests for materials should be addressed to Laura Balzer.

**Acknowledgments:**
We acknowledge the help of M. Pérez-Rodríguez, L. Sept, A. Prüßner and K. Braun during field work and P. Schmidt and A.
Calean for their help in sample preparation and laboratory analyses.

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

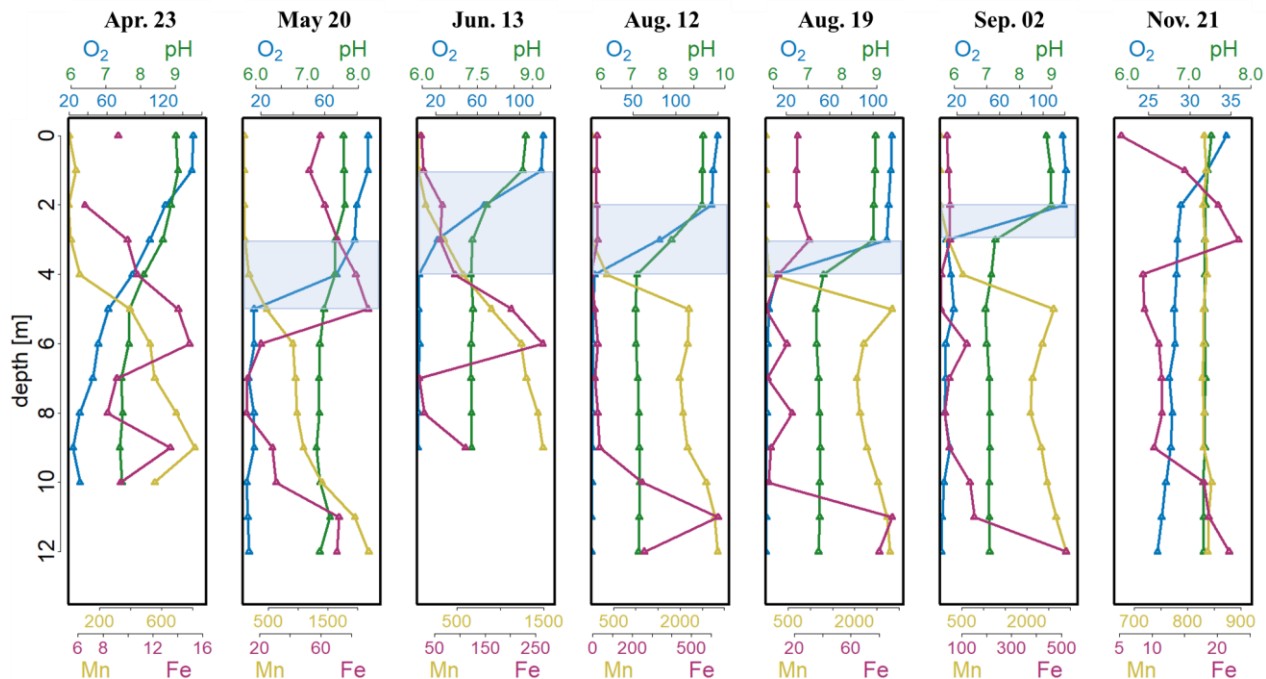


**Fig. 1.** Depth profiles of $O_2$ saturation (%), pH and concentrations of dissolved Mn [µg L$^{-1}$] and Fe [µg L$^{-1}$] in the Lake Ölper water column from April to November 2019. The depth of the sharp decrease in $O_2$ concentration and start of Mn reduction (RTZ) are shown in each panel (shaded light blue). Note that each column has an individual scale to better illustrate changes with depth. Depth profiles with the same scale in all columns are shown in the supplements (Fig. S1).






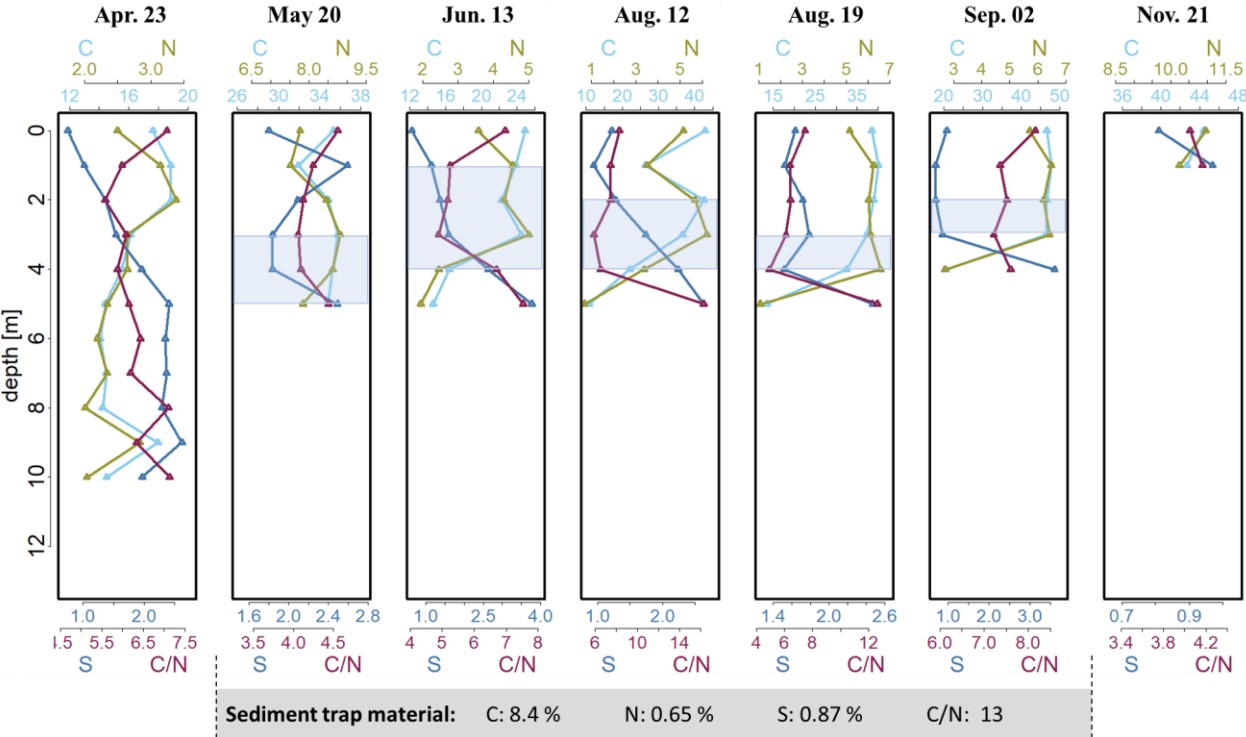

**Fig. 2.** Depth profiles of C [%], N [%], S [%] concentrations and calculated C/N ratio in the seston in Lake Ölper from April to November 2019. The depth of the sharp decrease in $O_2$ concentration and start of Mn reduction (RTZ) are shown in each panel (shaded light blue). Concentrations of C [%], N [%], S [%] and the C/N ratio of the sediment trap material collected

during the 141 days between May 6th and September 24th are given in the grey box below. Note that each column has an individual scale to better illustrate changes with depth. Depth profiles with the same scale in all columns are shown in the supplements (Fig. S4).





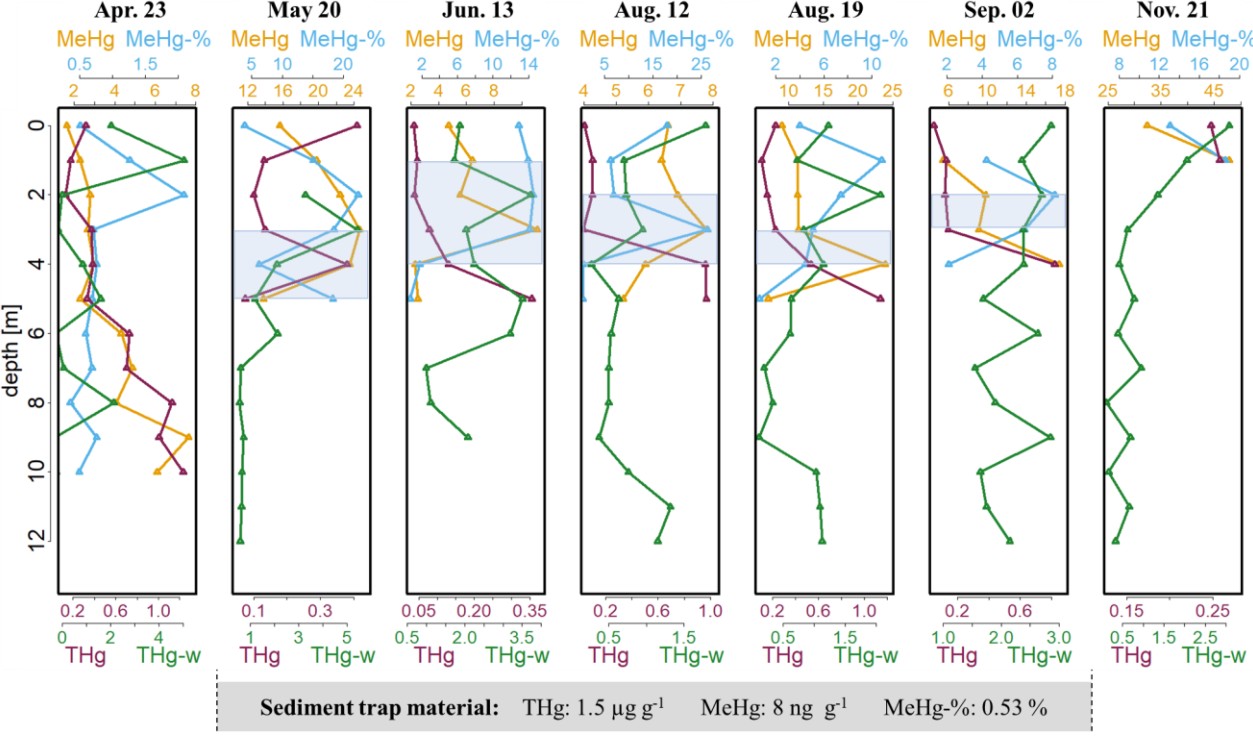

**Fig. 3.** Depth profiles of MeHg [ng g$^{-1}$] concentrations, percentage of MeHg [%] of THg (MeHg-%), THg concentrations in seston [μg g$^{-1}$] and dissolved THg [ng L$^{-1}$] in the water column (THg-w) of lake Ölper from April to November 2019. The depth of the sharp decrease in O$_2$ concentration and start of Mn reduction (RTZ) are shown in each panel (shaded light blue). Concentrations of THg [μg g$^{-1}$], MeHg [ng g$^{-1}$], MeHg-% [%] of the sediment trap material collected over the 141 days between May 6$^{th}$ and September 24$^{th}$ are given in the grey box below. Note that each column has an individual scale to better illustrate changes with depth. Depth profiles with the same scale in all columns are shown in the supplements (Fig. S5).



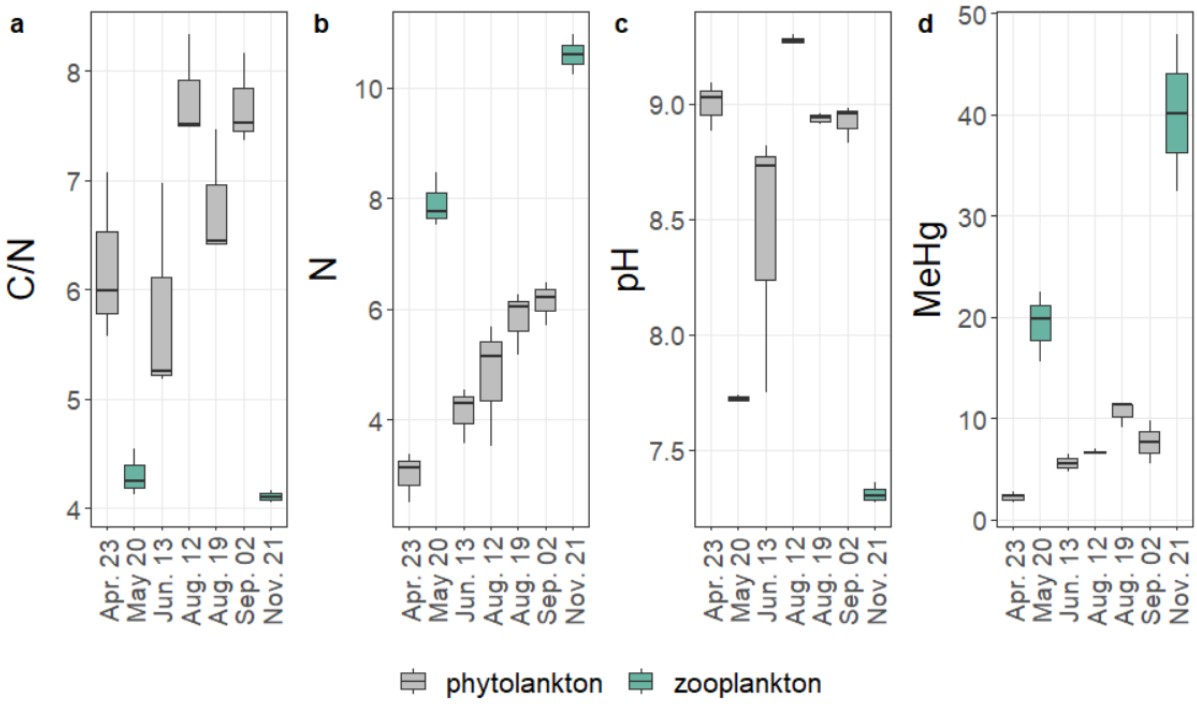

**Fig. 4.** C/N ratio (a), N concentration [%] (b), pH (c) and MeHg concentration [ng g$^{-1}$] (d) of the individual sampling days from the upper two oxic metres (0; 1; 2 m). Colours indicate differences in plankton dominance; green symbolizes higher amounts of zooplankton; grey symbolizes higher amounts of phytoplankton. The relative dominance of zooplankton was estimated from visible inspection and from pH only.