# Peer review of "Role of formation and decay of seston organic matter for the fate of methylmercury within the water column of a eutrophic lake"

_Biogeosciences, 2022_

## Author Response (AR1)

**Reply to reviewer comments on mansucript bg-2022-170**

**"Endogenic methylmercury in a eutrophic lake during the formation and decay of seston"**

**by Laura Balzer et al.,**

**Biogeosciences Discussion**

We thank the two reviewers for their constructive and helpful comments.

**Reviewer 1**

The paper aims to assess the role of seston in the production of methylmercury in a eutrophic lake. The paper is based on water, seston, trap and sediment sampling at seven dates between April and November 2019.

The paper is well written and structured, methods thoroughly described, and results generally well presented. In general, I like this paper, but I find that there are weaknesses in the design of the research and then over-interpretation of some results. MeHg has not been measured in the dissolved phase, then there is no clear picture of the overall situation. Based on partition coefficients, and on the concentration of seston in water column, it appears that usually most the MeHg in the raw water column is in the dissolved (or colloidal phase) (see Gallorini and Loizeau 2022, Chemosphere).

> *We thank the reviewer for this positive assessment.*
> *We are aware that analyses of MeHg in the dissolved phase would have been useful. However, our paper is focused on seston and specifically the fate of MeHg during decay of algae derived organic matter. Our intend was not to resolve the entire biogeochemical Hg/MeHg cycle in this lake. We further believe, that the situation is different in eutrophic lakes compared to oligotrophic lakes (such as lake Geneva in Gallorini and Loizeau 2022) regarding the partition of MeHg between the dissolved and the solid phase as there is so much more organic matter present during algae blooms that the dissolved phase is of minor importance here. Previous studies have shown, that dissolved Hg is depleted after algae blooms due to water phase Hg scavenging by sinking seston (Schütze et al, 2021).*

The authors rule out the possibility of diffusion of MeHg from the sediment, without clear evidence, as there is no measurement of seston and MeHg in the water column below RTZ. They invoke "that mixing is minor during times of high productivity (line 265)", however diffusion seems to occur and shown by the Mn profiles. The "pronounced" maximum concentration at the RTZ that is at the base of the all discussion and interpretation is not so evident. In line 250 it reads "During periods in which the RTZ was clearly defined, MeHg concentrations in seston showed a pronounced maximum at the RTZ that did not occur in April, when no RTZ was observed (Fig. 3)." The pronounced maximum is clear only in Aug. 19.

> *We cannot exactly follow the reviewer here, Fig. 3 shows that the maximum of MeHg concentrations is within the RTZ or directly below in May (there are two maxima) June, August and September. We added "directly below". The reason why there is no data from below the RTZ in some of the profiles is that there was not enough suspended matter below the RTZ which could be sampled with our method (25 μm net several 2 hours pumping) (see L254). From our data in the solid phase, we assumed that MeHg diffussion from the sediment is unlikely, but we agree with the reviewer that we cannot rule this out. We have now added the depth profiles of DOC from the different sampling days. MeHg released from bottom sediments is most likely bound to DOM as chloride concentrations in lakes are too low to be*

*competetive. The DOC profiles clearly indicate that DOC release from the sediment occurs as indicated by the highest DOC concentrations found in the deepest water samples and it is likey that MeHg released from decaying organic matter in the uppermost sediment layers is bound to DOM and distributed in the water column during lake mixing. However, DOC profiles do not show diffussion gradients during the summer months when the algae blooms occur and concentrations were even higher in the upper water layers indicating DOC release from decomposing algae organic matter which suggests rather MeHg formation in the water phase (labile algae derived DOM supports microbial MeHg formation in the water phase than uptake of MeHg released from the sediment although both is possible. We have revised the manuscript accordingly.*

Some discussed variations are very small and probably within uncertainties (e.g. C/N ratio). But the authors do not give uncertainty of the measurements, so it is impossible to evaluate the relevance of the variations.

*We have added uncertainities of the measurements, which, however, cannot explain the observed variations and trends in the data. We have now focused on the changes in the RTZ.*

Detailed remarks
L25 "The methylation of inorganic divalent forms of Hg(Hg(II)) to toxic MeHg is carried out…". The sentence implies that Hg(II) is not toxic, which is not the case.

*We agree. Has been changed accordingly.*

L 29 " being influenced by temporal and spatial variabilities". It is not clear to which processes these variabilities refer.

*We refer to the variability in redox-conditions. Has been revised.*

L 32 all these references on marine environment (19) are too much. Better to select the most relevant for your purpose.

*Has been changed accordingly.*

L50 About MeHg formation in lake snow, see Gallorini and Loizeau 2022, Chemosphere.

*Reference has been added.*

L84 As a pump and tubing have been used to sample water and seston, how potential contaminations (mainly for disssolved THg in water) have been evaluated?

*Pump and tubing has been cleaned (acid washed) thoroughly. Blanks were added.*

L89 text reads "PE Falcon tubes for." The end of the sentence is missing.

*Has been corrected.*

L94 text reads "but in most cases covered the upper 4 m". However, most figures indicate that the lower sample is at 5 m depth. Moreover, it is not clear why samples below 5 m were not collected.

*Has beenchanged to 5 m.*
*We were not able to gain sufficient material for solid phase analyses during 2 h sampling/ pumping as the amount of suspended matter below 5 m was, in most cases, very low. Longer pumping was not possible due to overheating of the pumps etc.*

L97 Is electrical conductivity corrected for temperature? Explain how it and other parameters were measured? From CTD or on the boat?

*Water parameters have been measured on the boat and are corrected for temperature.*

L99 The exposure time of the sediment trap (141 days) is very long, and then the material experience early diagenesis if no preservative was added. Then it is not clear why this sample was collected.

*The idea of the sediment-trap approach was to get an idea what the integrated material looks like after a period of some months of decomposition and if/how it differs from bottom sediments. A comment has been added.*

L162 change "the" to "then"

*Has been changed.*

L182 and following. It should be better stressed how the author link parameters to productivity. For instance, L188 text reads "Chlorophyll a concentrations were 2.5 to 2.8 8 μg, indicative of low phytoplankton productivity." Chl a is not a measurement of the productivity, as other factors may change the Chla concentrations (for instance grazing). Chla may be a direct proxy of algal biomass, not productivity.

*Has been changed to algal biomass production Consistently throughout the manuscript.*

L202 the profile of Fe in May is strange, as Fe(III) is essentially insoluble. So what is the "dissolved" species found in the upper layer in May? Then what happened in August 19, Fe dropped from > 500 to 100 ug/L and increase again > 500 in September.

*We did not analyse dissolved Fe-species, but we assume that the small amount of dissolved Fe found in the upper water layers is organically bound Fe, probably release during algae matter decay or from zooplankton. DOM-Fe is soluble under oxic conditions. The appearance of dissolved (reduced) Fe changes in the deep water layers between August 12, 19 and Sep.02 are most likely due to a change in redox zonation caused by more or less amount of suspended organic matter and differences in productivity/amount of algae biomass produced. Note, that pH is higher on Aug.12 = higher productivity compared to August 19. Similar, pH and Chl a at Sept 02 is higher (higher productivity) than on Aug. 19.*

L203 text reads "After mixing in November, the Mn and Fe concentrations were uniformly low". From Fig 1, Mn isn't low in November, with values much higher than in surface waters measured the other months.

*We agree, although the message is clear that there is no more redox zonation. Text has been changed.*

L207 C/N ratio compare organic carbon to organic nitrogen in samples. Is all C in the sample from organic matter? For instance, the sediment trap results indicate C concentration of ~9%, that is 18 to max 30% of the sediment is organic matter. What is the composition of the remaining 70% of the sediment? Does it contain C as carbonates? This point should be clarified.

*Has been clarified. The remaining material in the traps is mineral matter we assume that also considerable amount of biogenic silica derived from diatoms could be found in the trap material, but this has not been analysed. There is no carbonate formation in the lake. In addition, samples have been decarbonated prior to carbon analysis. In addition, in the deep layers where the sediment trap was installed the lower pH will cause dissolution of calcite.*

L213 The decrease of C/N ratio explained by mineralization is not obvious. A reference is needed here, as usually it is the reverse that is observed as mentioned the given reference Meyers and Lallier Vergès 1999. Moreover, is the decomposition the only processes, what about selected grazing or change in composition of the seston (phyto vs. zooplankton) to explain the C/N variation?

> *We agree with the reviewer that changes in C/N ratio above the RTZ are probably too small to undoubtedly indicate organic matter decomposition. We also agree, that some of the small changes seen here could have been caused by the occurrence of zooplankton. We have therefore restricted the interpretation of C/N ratios as a measure for organic matter decomposition to the values within or below the RTZ.*

L260 text reads "This would explain the continuous increase in MeHg concentrations with depth…" What is the explanation? The absence of microniches does not explain the formation of MeHg at depth, where O2 saturation is still > 20%. Diffusion from sediments?

> *Sentence has been changed. We believe that the increase in MeHg concentrations with depth in April is mainly caused by mass loss due to progressive organic matter decomposition (comparable to what has been described by Gallorini et al, 2022) although we cannot exclude MeHg formation by Mn reducing bacteria or release of MeHg from the sediment (in April) and coupling to DOC. A clarifying sentence has been added.*

L294 Mass loss is the only explanation of the THg increase with depth. However, C concentration decrease by max a factor 3.9, whereas THg increase is a factor 26. Then the mass loss cannot account totally for the increase in THg concentration.

> *We are not sure if we understand this comment correctly. THg and C do not necessarily have to increase by the same exent because C (and other elements) is lost during mineralisation, but Hg is not. An additional explanation might be that some Hg released to the water phase during organic matter decomposition is scavenged by sinking seston as it has been observed in marine studies.*

L314 text reads "The sulphide produced may form insoluble complexes with Hg (Shanks and Reeder, 1993; Bianchi et al., 2018), such as Hg sulphides (HgS), meaning that Hg becomes less available for methylation" It is not so clear that the presence of S decreases the bioavailability of Hg. Barrouilhet et al 2022 ESPR show that methylation potential increases with S concentration before to decrease at high S concentration.

> *Our data indicates that there is no sulfate reduction and thus formation of sulphide. The increase in S concentration in sesteon is thus rather due to mass loss during organic matter decomposition. This was an assumption which we could not proof in the frame of this study (only based on the increase of S concentration). The amount of material gained was too small to do Hg-thermo-desorption analyses or similar. The study of Barrouilhet et al 2022 is quite different from what we did and we could hardly say if their findings do apply here.*

L318 text reads "Thus, THg and MeHg fluxes to the sediment are largely determined by changes in OM composition and mass loss during decomposition." While these processes may change the MeHg fluxes, it is not clear why these processes change the flux of THg: i) if the authors are correct, the increase in THg concentration is due to mass loss in OM, then the quantity of THg remain the same, so the flux, and transformation of THg to MeHg will not change the flux of THg as MeHg is included in THg.

> *We agree, sentence has been removed*

L341 MeHg concentrations in the sediments are not sufficient to assess fluxes from sediments to interstitial water to overlying water.

*We agree, statement has benn toned down.*

L354 "Water column MeHg formation and degradation in eutrophic lakes appears to be intense and occurs rapidly and at rates similar to what we observed within the bottom sediments" This statement is not supported by the data/discussion. No rate has been determined neither in the seston nor in sediments.

*Has been removed.*

Fig 1. The scales do not cover the entire range of the results.

*Has been adapted in all figures including the figures in the supplements.*

**Reviewer 2**

General comments.
This paper builds on several prior studies that show that the water column of lakes and oceans can be an important site for MeHg formation. It differs from most water column studies by focusing on a eutrophic urban lake and by specifically targeting MeHg abundance in bulk seston at different depths and dates for clues about formation and decay mechanisms. Unfortunately, the sampling technique lumped zooplankton in with seston, potentially introducing bias due to biomagnification. And the sampling scheme was spatially inconsistent, which makes the comparison of depth profiles on different dates difficult. The reason that the entire water column was sampled on one date and only the upper water column on most other dates is unexplained, and it compromises the authors' conclusions about what's going on as particles sink (especially in the hypolimnion since it was rarely sampled). Among other things (below), the authors need to justify their sampling methods and revisit the interpretation of changes in Hg speciation across depth and time. They also need to reconsider conclusions about links between climate change, productivity and bioaccumulation. This will require major revision.

> *Our focus was on bulk seston and to our knowledge zooplankton is part of seston (: minute material moving in water and including both living organisms (such as plankton and nekton) and nonliving matter (such as plant debris or suspended soil particles).) We also believe, that the distribution of zooplankton alone cannot explain the THg and MeHg depth profiles in our lake. The separation of phyto- and zooplankton is useful in studies on biomagnification, which was not our topic. In this case, a qualitative separation of both fraction in small amounts is sufficient. In case of bulk seston sampling, eg. by means of a pump-and-sieve/filter system (0.45 µm?) during algae blooms as suggested by the reviewer such qualitative separation is nearly impossible (agglomeration) if larger sample volume is needed. Moreover, we believe that our interpretation of the distribution of MeHg and THg in the water phase is supported by our data on algae biomass, (Chl 1, pH) is sound. We regret that our sampling was imperfect, we made a lot of effort to gain in all cases sufficient material, however, this is a natural system with sometimes unpredictable changes of conditions. The reason why sampling is inconsistent through time and space is that we could not get sufficient material from the hypolimnion within the possible pumping time (~ 2 h per layer). We already mentioned this in the text but we have explained this in more detail (see L254).*
> *We have tone down on our conclusion regarding the link between climate change, productivity and bioaccumulation. We assume that the reviewer based this comments on his assumption that we mainly see biodilution. We have commented on this below.*

Specific comments.
The term "endogenic" should be reconsidered. It means "within the system", which for lakes technically includes sediments. "Water column" would be better, unless they mean "within the seston" – in which case the title and text need to be re-worded

> *This term has been introduced in other studies. For example, in Gallorini et al. 2021: We have defined it within the manuscript as "production within the water column".*

Line 89 is an incomplete sentence

> *Has been changed.*

Line 90: why a 25µm net? It would allow many cyanophytes and chlorophytes to pass through, and bias collection toward zooplankton (which are not "seston"). Why not a clean pump-and-sieve/filter system instead?

> *We agree with the reviewer that some of the small fraction of phytoplankton might have got lost during water pumping through a 25 µm net. We tried a pump-and –sieve filter system*

*before. However, this took too long to gain sufficient material from each water layer to do the solid phase analyses needed here at a resolution of 1 m within a single day (filter clogging etc., batteries etc.). Because of this, we decided to pump the water through a 25 μm net. Although it would have been the best option to sample all phytoplankton fraction, we believe that the lack of the fraction < 25 μm has no significant influence of the overall results and conclusions of this study (it just means more phytoplankton). To our knowledge zooplankton is part of the seston, too.*

L220-225. The seston samples collected on those dates are not really much closer to the sediment surface. There's just one hypo sample and it's directly beneath the RTZ.
You'd need to sample more depths to justify. Revise.

> *The reviewer is right the deepest sample was taken just below the RTZ in those months. We have revised this statement. As explained above. We were not able to gain sufficient material from deeper layers in those month within the possible pumping time >2 h).*

L235. But peak concentrations of MeHg in seston occur in the suboxic RTZ on 4 of the 5 dates when the lake was strongly stratified. On the remaining date, seston MeHg concentrations are highest in the upper hypolimnion. During stratification, MeHg is never highest in the oxic epilimnion.

> *We agree that this sentence is misleading and have clarified this section.*

If anything, these finding suggest that MeHg production is associated with microbial respiratory pathways that are less energy efficient than $O_2$ reduction (e.g. sulfate reduction, Fe reduction). Revise.

> *Not clear what the reviewer means here and what should be revised. We discussed in the ms that MeHg formation appears to be releated to redox conditions in the water column specifically to Mn reduction (similar to what has been shown by Petersen et. al 2020(in a lake) and by Kohler et la., 2022 (in the arctic ocean) Fe reduction is of minor importance in this lake and sulphate reduction does not take place in the water column (compare Fig. S3). Data on eutrophic lakes are rare and to our knowledge not available yet at similar high resolution. A major aim of this study is to show changes in MeHg in seston at this comparatively high temporal and spatial (depth) resolution to understand the evolution of MeHg and THg concentrations and proportion during sinking through the water column.*

L240-245. Alternatively, low MeHg during high productivity may reflect biodilution in the larger phytoplankton biomass (i.e. parental seston). Lacking sound data, one can't distinguish zooplankton bias from biodilution in microplankton, and neither necessarily point to sestonic microniches. Revise

> *We have revised the section on the variation of MeHg concentrations and the influence of changes in algal biomass production. Our data clearly indicates that MeHg concentrations are rather positvely related to algal biomass production and not negatively as expected in case of biodilution, so that rather redox conditions than biodilution could explain the observed variability in MeHg concentrations.*
> *The following paragraph has been added.*
>
> *L 260-: „ It has been reported mainly based on laboratory experiments that MeHg concentration in algae material are lower when algal biomass is high (biodilution) (Chen and Folt, 2005; Pickhardt et al., 2002). However, biodilution seems to be of minor importance in lake Ölper. In April, when no RTZ was established, MeHg concentrations showed a continuous increase with depth as $O_2$ concentrations decreased and had a comparatively low maximum of 7.6 ng g-1 at 9 m depth (Fig. 3). Biodilution might explain the simultaneous increase in MeHg and THg concentrations and the decrease in algal biomass (Chla) with*

*depth here, where redox zonation was not pronounced throughout the entire water column (Fig. S2). However, we assume that the observed increase in MeHg and THg concentrations with depth in April are rather caused by mass loss in the sinking seston during decomposition and decreasing redox potential than by biodilution in the epilimnion. Furthermore, September has the highest algal biomass (indicated by the highest Chla concentration) but does not have the lowest MeHg concentrations. In contrast, September has higher MeHg concentration than April, June and August 12. These high MeHg concentrations could not be explained by abundance of high amounts of zooplankton as relatively high C/N ratios (compared to May and November) indicate that the seston here is dominated by algal OM. Similar, algal biomass production (Chla) is relatively lower in June and so are MeHg and THg concentrations. This positive relationship between algal biomass and MeHg as well as THg concentrations clearly indicates that biodilution could not explain the observed spatial and temporal variation in MeHg and THg concentrations in our lake."*

L255-263. They could also be explained by the presence of free-water microbes that possess the methylation gene pair hgcAB and occupy the O/A boundary. DOM rather than POM could be their carbon source. Revise.

*We discussed this point in L 266. It is likely that free-water microbial Hg methylation occurs, specifically because there is predominantly easy accessible DOM in the water column. But to our knowledge, oxic microbial pathways of MeHg formation are not yet known. Many papers point to methylation within anoxic microniches. However, our data suggest that free-water microbial Hg methylation is rather not the dominant process here as high MeHg concentration only occur during times of a pronounced RTZ (compare April when production is already high but MeHg is low because redox-zonation is not yet established) We tried to make this point clearer in the ms.*

L275-284. Sestonic MeHg in the 20% range is not atypical for unpolluted temperate lakes. What's unusual is the very low %MeHg in April

*We can tone this statement down. However, we think that 20 % is a lot regarding the high biomass and that there is no influence from soil derived DOC/MeHg-rich inflow, which is typical for many oligotrophic lakes.*
*The low MeHg proportions and concentrations in April (compared to the summer months) supports our conclusion that the MeHg is predominantly formed in the water phase along redox-gradients/micro-niches and the role of the RTZ. In April, algae biomass and productivity as indicated by high Chl a and high pH is already high, but the redox gradients are only weakly developed (only weak Mn-reduction) (Fig. S1), most likely because organic matter decomposition in the water column is still low. If most MeHg is originated from the sediment, we should see this MeHg in the seston also in April.*
*We tried to make this point clearer in the ms.*

L346. Actually, this was first shown in Little Rock Lake, which is only 10m deep (but the eutrophic part may be right).

*We have referred to this point in the ms. However, we actually pointed out that our study is focused on eutrophic lakes, where data is rare.*

L346-end. Note that the range of Hg and MeHg in the seston of this eutrophic lake is on the low end of seston data reported for mesotrophic to oligotrophic North American lakes, both for MeHg concentration and %MeHg. High productivity is not necessarily conducive to abnormally high rates of MeHg accumulation in bioseston. In fact, most data suggest the opposite due to biodilution. It may be true that higher amounts of OM decomposition in eutrophic lakes does indeed exacerbate O2 depletion

and enhance methylation in suboxic water, but that was not measured here. It seems that the most you can say with the data presented here is that the opposing forces of high biodilution and high decomposition need to be reconciled before addressing the impact of climate change. Revise

> *We don't think that our data is directly comparable to the studies of* mesotrophic to oligotrophic North American lakes *mentioned by the reviewer, because most of these lakes have influx of MeHg from their catchment ad they are not eutrophic. Our aim here was to show the spatial and temporal changes in MeHg and THg concentration in seston during algae blooms, but we agree that the proportion of MeHg found in seston of lake Ölper might not be exceptionally high. How many studies on North American lakes show the distribution of THg and MeHg in seston at high temporal and spatial resolution including redox conditions, Chl a data etc.? Moreover, we do not agree with the reviewer that our data could be explained by biodilution in contrast our data suggest that biodilution is of minor importance here and cannot explain the observed variability in MeHg concentrations. Please see comments above and explanation in the revised manuscript.*

It may be true that higher amounts of OM decomposition in eutrophic lakes does indeed exacerbate O2 depletion and enhance methylation in suboxic water, but that was not measured here.

> *We believe that this is exactly what we have measured in our study.*

---

## Referee Report (RR1)

Review of Balzer et al R1. See comments in red below.

My recommendation for R1: Revisions inadequate. Reject

Referee Review of Balzer et al. (2022): "Endogenic mercury…"

General comments.

This paper builds on several prior studies that show that the water column of lakes and oceans can be an important site for MeHg formation. It differs from most water column studies by focusing on a eutrophic urban lake and by specifically targeting MeHg abundance in bulk seston at different depths and dates for clues about formation and decay mechanisms. Unfortunately, the sampling technique lumped zooplankton in with seston, potentially introducing bias due to biomagnification. And the sampling scheme was also spatially inconsistent, which makes the comparison of depth profiles on different dates difficult. The reason that the entire water column was sampled on one date and only the upper water column on most other dates is unexplained, and it compromises the authors' conclusions about what's going on as particles sink (especially in the hypolimnion since it was rarely sampled). Among other things (below), the authors need to justify their sampling methods and revisit the interpretation of changes in Hg speciation across depth and time. They also need to reconsider conclusions about links between climate change, productivity and bioaccumulation. This will require major revision.

Most of the issues raised above remain unresolved in the revised MS. The reason(s) that they sampled only the upper water column on most dates have not been given; and, in contradiction, they claim to have sampled the entire water column on 7 dates at 1m depth intervals. The interpretation of changes in Hg speciation across space and time continues to be largely speculative.

Specific comments.

1. The term "endogenic" should be reconsidered. It means "within the system", which for lakes technically includes sediments. "Water column" would be better, unless they mean "within the seston" – in which case the title and text need to be re-worded. This term remains problematic. The authors refer to anoxic microniches in sinking particles as important sites for MeHg formation (e.g. "lake snow"), but the collection and analytical methods don't target these zones in dead suspended aggreagtes. Instead, they target live plankton that acquire MeHg by absorption or ingestion. That can't tell us anything about MeHg formation pathways in anoxic microniches or anywhere else in the lake .

2. Line 89 is an incomplete sentence

3. Line 90: why a 25um net? It would allow many cyanophytes and chlorophytes to pass through, and bias collection toward zooplankton (which are not "seston"). Why not a clean pump-and-sieve/filter system instead? R1 L109:This question has not been resolved and it is a fatal flaw. "Seston" is the nonliving particulate matter in the water column, as opposed to "plankton" which is the live phytoplankton, nanoplankton and zooplankton.

The sample collection  method used in this paper would be strongly biased toward plankton. As living organisms, plankton do not have the anoxic microniches (generally attributed to "lake snow".) Instead they often have defense mechanisms that prevent the accumulation of microbes on their surface. In short, this paper does not directly address anything about anoxic microniches in lakes.

4. L220-225. The seston samples collected on those dates are not really much closer to the sediment surface. There's just one hypo sample and it's directly beneath the RTZ. You'd need to sample more depths to justify. Revise. R1 L245-250. The small number of samples is still a serious limitation. Obviously, O2 depletion indicates that high rates of metabolically efficient decomposition have occurred. That's why there is an RTZ and anoxic hypo in the first place. There are no further insights into MeHg formation or demethylation in the data.

5. L235. But peak concentrations of MeHg in seston occur in the suboxic RTZ on 4 of the 5 dates when the lake was strongly stratified. On the remaining date, seston MeHg concentrations are highest in the upper hypolimnion. During stratification, MeHg is never highest in the oxic epilimnion. If anything, these finding suggest that MeHg production is associated with microbial respiratory pathways that are less energy efficient than O2 reduction (e.g. sulfate reduction, Fe reduction). Revise. R1 L264-267. The revised text is better( more aligned with the data), but it now argues against their premise that methylation is occurring mainly in anoxic microniches within decaying seston as it settles. When the classic redox sequence has set up in the water column, the anaerobic microbes that possess the hgcAB genes can produce MeHg, but they don't have to reside within anoxic microniches in settling POM. Nothing in the data presented in this paper indicates or proves that they do. Instead, Mn, Fe and SO4 reducers may simply set up shop at the optimum depth and utilize the flow of nutrients and terminal electron acceptors from above (or below). This may occur in sediments or the water column. None of this is news, and none of it necessarily involves anoxic microniches in seston.

6. L240-245. Alternatively, low MeHg during high productivity may reflect biodilution in the larger phytoplankton biomass (i.e. parental seston). Lacking sound data, one can't distinguish  zooplankton bias from biodilution in microplankton, and neither necessarily point to sestonic microniches. Revise. I'm not convinced by the arguments in R1.

7. L255-263. They could also be explained by the presence of free-water microbes that possess the methylation gene pair hgcAB and occupy a region below the O/A boundary. DOM rather than POM could be their carbon source. Revise. R1 L308-324. This section remains highly speculative, and absent more rigorous investigation, alternative hypotheses can't be evaluated. The authors "assumption" that their explanation is the correct one isn't convincing

8. L275-284. Sestonic MeHg in the 20% range is not atypical for unpolluted temperate lakes. What's unusual is the very low %MeHg in April. R1. L350-358. The mention of O2 fluxes into settling particles again assumes we are dealing with nonliving POM, but

it's more likely that the "seston" collected in the plankton net comprises live organisms. The conflation of plankton with "lake snow" or dead POM aggregates is a conceptual problem throughout this paper

9. L346. Actually, this was first shown in Little Rock Lake, which is only 10m deep, and subsequently in many other lakes in this depth range. Not just deep oligotrophic lakes(but the eutrophic part may be right). R1. L346-347. This has not been addressed. The text remains unchanged and it seems like an attempt to oversell the novelty of this research.

10. L346-end. Note that the range of Hg and MeHg in the seston of this eutrophic lake is on the low end of seston data reported for mesotrophic to oligotrophic North American lakes, both for MeHg concentration and %MeHg. High productivity is not necessarily conducive to abnormally high rates of MeHg accumulation in bioseston. In fact, most data suggest the opposite due to biodilution. It may be true that higher amounts of OM decomposition in eutrophic lakes does indeed exacerbate O2 depletion and enhance methylation in suboxic water, but that was not measured here. It seems that the most you can say with the data presented here is that the opposing forces of high biodilution and high decomposition need to be reconciled before addressing the impact of climate change. Revise R1 L349-357. This text also remains unchanged and it continues to promote the importance of anoxic microniches despite the fact that there is no direct evidence. The authors make claims about rates of methylation and demethylation without any rate determinations. Entirely unsupported speculations.

---

## Author Response (AR2)

Dr Sebastian Naeher
Associate Editor, Biogeosciences
Biogeosciences

Dear Dr. Naeher,

**Subject: Submission of revised manuscript bg-2022-170; "Endogenic methylmercury in a eutrophic lake during the formation and decay of seston" by Laura Balzer et al.**

Thank you for handling the manuscript.
We have carefully revied the comments of the reviewers. Our responses are given in a point-by-point manner below. Changes in the manuscript are shown in a marked-up version of the manuscript.
The reviewers commented on our sampling technique and reviwer2 is concerned about the term "seston" and our sample content. According to him we have sampled only living plankton (with a 25µm plankton net) and he believes, that the term "seston" only comprises non-living particulate matter. We give a detailed point-by-point response below why the think that this is not correct and why we still want to use the term seston in our manuscript. In short:
The use of plankton nets to sample seston is quite common in the literature (e.g. Yigiterhan et al., 2020, Biogeosciences).
In all references we found, including text books, seston is defined as all particles suspended in water regardless of their nature or origin including both, living plankton (phytoplankton, zooplankton, bacterioplankton, pseudoplankton, paraplankton) and detritus (dead biogenous, terrigenous, aerogenous and anthropogenous material).
Our seston samples contain a varying composition of plankton and detritus, with higher amounts of living plankton at the surface. Seston is therefore the best superordinate term to describe all our samples from the surface to the hypolimnion. The obvious change in colour of our samples from light green to light brown clearly indicates that the material does not only consist of living phyto- and zooplankton. Instead, the changing colour is mainly indicating decaying organic matter. If our samples would include only living biota all detritus must be < 25 µm which is unlikely. Moreover, if we do not see decay in our material, as suggested by reviewer2 we would not see oxygen depletion and formation of a RTZ.
The main idea of our study is to show how THg and MeHg concentrations and proportions change during organic matter decay in sinking seston in the water phase during algae blooms and this approach is new, to our knowledge. We suggested the formation of MeHg in seston micro niches as a possible process to explain the high methyl-Hg proportions in the samples of the upper water layers but we agree with the reviewer that we did not show this and it is not our point.

We have included a more detailed explanation why we have taken the samples in this way and a definition of our seston samples in our manuscript.
We have also replaced the term endogenic to "within the water column" or water column, respectively and modified the title of the manuscript which now reads "Role of formation and decay of seston organic matter for the fate of methylmercury within the water column of a eutrophic lake".

We hope that the revised version is now suitable for publication and look forward to hearing from you in the near future

Sincerely,

Laura Balzer

**Review 2**

I understand that the aim of the paper is not to resolve the entire Hg cycle in the lake, but a focus on the role of the seston in Hg/MeHg. But then the sampling strategy to collect the seston with a 25 µm net excludes a portion of the seston. What proportion of the seston present in the water column is not sampled and analysed? An assessment of the implications of this partial(?) sampling would strengthen the interpretation of the results.

We agree with the reviewer that this sampling strategy of only sampling particles >25 µm is not so common in Hg analyses. All proportions of seston that are smaller 25 µm were not sampled (nano and pico plankton/particles).

Our goal was to study the temporal and spatial occurrences of lacustrine MeHg in settling particles at high resolution and how it changes during OM decomposition throughout the water column. Therefore, we needed sufficient material from each water layer to do the solid phase analyses to be able to show changes within 1 m of the water column. Sediment traps, for example, collect settling particles as a bulk sample integrating sedimentation over a specific time (days, weeks..) and the whole water column above. By sampling at one day in 1m interval directly from the water column we were able to cover daily lake fluctuations like stratifications that may change from hours to weeks. (Ortiz et al., 2015) showed that "methylation can occur as long as large particulates are present (>8 µm) [and that] it is unlikely that conditions conducive to methylation would occur in the smallest size fraction, which is likely composed of individual particles, small phytoplankton, and other microbes". Ortiz et al. 2015 concluded that methylation by anaerobes in oxic waters must be due to the formation of reduced oxygen microzones within the larger aggregations. As we said before, we tried a pump-and –sieve filter system before. However, this took too long to gain sufficient material from each water layer to do the solid phase analyses needed here at a resolution of 1 m within a single day (filter clogging etc., batteries etc.). Because of this, we decided to pump the water through a 25 µm net. In this way we obtained material that contains larger aggregates which could provide an ideal environment for mercury methylation because of the formation of anaerobic conditions. Although it would have been the best option to sample all phytoplankton fraction, we believe that the lack of the fraction < 25 µm has no significant influence of the overall results and conclusions of this study. Although we agree with the reviewer that nano and pico-plankton is involved in MeHg uptake and alteration (see Cossart et al., 2021)

We will add the following text for clarification (L104..)

*"With the used method it was not possible to gain sufficient material from deeper layers in the water column, as the amount of suspended matter below 4-5 m was, in most cases, very low. [All seston samples were frozen immediately after sampling and subsequently freeze-dried and homogenized with a glass pestle for further analyses (THg, MeHg, CNS).]*

*Herein, we define seston as all particles suspended in the water column, including plankton (phytoplankton, zooplankton, bacterioplankton, pseudoplankton, paraplankton) and detritus (biogenous, terrigenous, aerogenous and anthropogenous detritus) (Lenz, 1977), larger in size than 25 µm. This method does not distinguish between the two types of seston nor*

*further between phyto- and zooplankton. Thus, our seston samples are a collection of varying compositions of plankton and detritus and their subgroups.*

*This approach excluded the pico- and nano-sized seston fraction (< 25 μm). We are aware that the smaller fraction is of importance within the microbial loop and would potentially extend our data. Pumping the water through the 25 μm net was the best method and within the range of our possibilities, which provided enough material for all solid analyses and allowed us a high sampling frequency (each water layer at a resolution of 1 m within a single day at several days a year) in the best of our abilities.Ortiz et al. (2015) showed that anoxic microinches can be formed within aggregations as long as the particles are larger than 8 μm. Smaller particles (composed of individual particles, small phytoplankton, and other microbes) do not provide ideal conditions for Hg methylation as no anoxic microniches can occur (Ortiz et al., 2015). Thus, we are confident, that seston >25μm size allow us to study the temporal and spatial occurrences of lacustrine MeHg in settling particles, how it changes during OM decomposition throughout the water column, to cover daily lake fluctuations like stratifications that may change from hours to weeks and to analyse if anoxic microniches may be formed also in shallow eutrophic lakes."*

**Review 1**

Review of Balzer et al R1. See comments in red below. My recommendation for R1: Revisions inadequate. Reject Referee Review of Balzer et al. (2022): "Endogenic mercury…" General comments.

This paper builds on several prior studies that show that the water column of lakes and oceans can be an important site for MeHg formation. It differs from most water column studies by focusing on a eutrophic urban lake and by specifically targeting MeHg abundance in bulk seston at different depths and dates for clues about formation and decay mechanisms. Unfortunately, the sampling technique lumped zooplankton in with seston, potentially introducing bias due to biomagnification. And the sampling scheme was also spatially inconsistent, which makes the comparison of depth profiles on different dates difficult. The reason that the entire water column was sampled on one date and only the upper water column on most other dates is unexplained, and it compromises the authors' conclusions about what's going on as particles sink (especially in the hypolimnion since it was rarely sampled). Among other things (below), the authors need to justify their sampling methods and revisit the interpretation of changes in Hg speciation across depth and time. They also need to reconsider conclusions about links between climate change, productivity and bioaccumulation. This will require major revision.

Most of the issues raised above remain unresolved in the revised MS. The reason(s) that they sampled only the upper water column on most dates have not been given; and, in contradiction, they claim to have sampled the entire water column on 7 dates at 1m depth intervals. The interpretation of changes in Hg speciation across space and time continues to be largely speculative.

The reason why there is no data from below the RTZ in some of the profiles is that there was not enough suspended matter below the RTZ which could be sampled with our method (25 µm net several 2 hours pumping) (see L101-103).

We will add the following sentence: *"With the used method it was not possible to gain sufficient material from deeper layers in the water column, as the amount of suspended matter below 4-5 m was, in most cases, very low."*

In L 91-94 of the revised MS we wrote " Water and seston samples were taken on seven days between April and November 2019 using a clean stainless steel immersion pump (Comet Combi 12–4T). The water column was sampled over the deepest (~12 m) portion of the lake. Samples were collected from the surface down to the sediment water interface at 1 m intervals." We will change the last sentence to "*water* samples were collected from…. " to make clear that only the water samples has been collected from the surface down to the sediment.

Specific comments.

1. The term "endogenic" should be reconsidered. It means "within the system", which for lakes technically includes sediments. "Water column" would be better, unless they mean "within the seston" – in which case the title and text need to be re-worded. This term remains problematic. The authors refer to anoxic microniches in sinking particles as important sites for MeHg formation (e.g. "lake snow"), but the collection and analytical methods don't target these zones in dead suspended aggreagtes. Instead, they target live plankton that acquire MeHg by absorption or ingestion. That can't tell us anything about MeHg formation pathways in anoxic microniches or anywhere else in the lake .

*We will change the term endogenic to "within the water column" or water column, respectively.*
*We will change the title „Endogenic methylmercury in a eutrophic lake during the formation and decay of seston" to:*
*"Role of formation and decay of seston organic matter for the fate of methylmercury within the water column of a eutrophic lake"*

*Regarding the term seston. See answer below.*

2. Line 89 is an incomplete sentence

Has been corrected

3. Line 90: why a 25um net? It would allow many cyanophytes and chlorophytes to pass through, and bias collection toward zooplankton (which are not "seston"). Why not a clean pump-and-sieve/filter system instead? R1 L109: This question has not been resolved and it is a fatal flaw. "Seston" is the nonliving particulate matter in the water column, as opposed to "plankton" which is the live phytoplankton, nanoplankton and zooplankton. The sample collection method used in this paper would be strongly biased toward plankton. As living organisms, plankton do not have the anoxic microniches (generally attributed to "lake snow".) Instead, they often have defence mechanisms that prevent the accumulation of

microbes on their surface. In short, this paper does not directly address anything about anoxic microniches in lakes.

In all references we found, including text books, seston is defined as "all particles suspended in water regardless of their nature or origin. Depending on the aspect being dealt with, the particles can be classified under different headings, for instance according to particle size or chemical composition" (Lenz, 1977). Here is written that Seston can include both, plankton (including phytoplanklton, zooplankton, bacterioplankton, pseudoplankton, paraplankton) and detritus (including biogenous, terrigenous, aerogenous and anthropogenous detritus). We decided to use the term seston to include all the mentioned types of plankton and detritus, as our sampling method did not distinguish between the two types of seston nor further between phyto- and zooplankton. The relative contributions of plankton and detritus and their subgroups can differ significantly between samples and is hard to distinguish (Yigiterhan et al., 2020).( "There have been few studies that tried to distinguish the relative contributions of biotic and abiotic particles in marine particulate matter (Lam et al., 2015; Ohnemus and Lam, 2015; Ohnemus et al.,2017; Wen-Hsuan Liao et al., 2017)" from (Yigiterhan et al., 2020).) We assume that the samples from the surface contain higher amounts of living plankton than samples from the RTZ and the hypolimnion that contain higher amounts of dead detritus and abiotic particles. But we have not analysed the specific amounts of plankton or detritus in our samples. Seston is therefore the best superordinate term to describe all our samples from the surface to the hypolimnion. The obvious change in colour of our samples from light green to light brown clearly indicates that the material does not only consist of living phyto- and zooplankton. Instead, the changing colour is mainly indicating decaying organic matter. If our samples would include only living biota all detritus must be < 25 µm which is unlikely. The use of plankton nets to sample seston is quite common in the literature (e.g. Yigiterhan et al., 2020, Biogeosciences). Moreover, if we do not see decay (but only live plankton) in our samples, as suggested by the reviewer we would not see oxygen depletion and formation of a RTZ.

We will include the following sentences for clarification:

*L 107-111"Herein, we define seston as all particles suspended in the water column, including plankton (phytoplankton, zooplankton, bacterioplankton, pseudoplankton, paraplankton) and detritus (biogenous, terrigenous, aerogenous and anthropogenous detritus) (Lenz, 1977), larger in size than 25 µm. This method does not distinguish between the two types of seston nor further between phyto- and zooplankton. Thus, our seston samples are a collection of varying compositions of plankton and detritus and their subgroups."*

Regarding the second point of the reviewer "anoxic microniches":

It was shown in previous studies that anaerobic conditions can be formed in the centre of marine snow even if the aggregation contain photosynthetic active organisms (Alldredge and Cohen, 1987; Shanks and Reeder, 1993). Based on this, even if our samples contain living phytoplankton it is possible that they agglomerate with other particles to larger aggregates that provide an ideal environment for mercury methylation because of the formation of anaerobic conditions (Alldredge and Cohen, 1987). The model of anoxic microniches includes

the formation of MeHg by microbes within/ in the centre of the aggregated particles and not on their surface as commented by the reviewer (Alldredge and Cohen, 1987).

We will include the following paragraph for clarification:

*"It was shown in previous studies that anaerobic conditions can be formed in the centre of marine snow even if the aggregation contain photosynthetic active organisms (Alldredge and Cohen, 1987; Shanks and Reeder, 1993). We did not distinguish the relative contributions of plankton and abiotic particles in our seston samples. We suggest that the seston samples from the surface contain higher amounts of living plankton than samples from the RTZ and the hypolimnion that contain higher amounts of dead detritus and abiotic particles as oxygen got depleted with depth. Anaerobic conditions are more likely to form the larger the particles and the less photosynthetically active the particles are (e.g. in the dark) (Alldredge and Cohen, 1987) "*

4. L220-225. The seston samples collected on those dates are not really much closer to the sediment surface. There's just one hypo sample and it's directly beneath the RTZ. You'd need to sample more depths to justify. Revise. R1 L245-250. The small number of samples is still a serious limitation. Obviously, O2 depletion indicates that high rates of metabolically efficient decomposition have occurred. That's why there is an RTZ and anoxic hypo in the first place. There are no further insights into MeHg formation or demethylation in the data.

See comment above.

The main idea of our study is to show how THg and MeHg concentrations and proportions change during organic matter decay in sinking seston in the water phase during algae blooms and this approach is new, to our knowledge. We suggest the formation of MeHg in seston microniches as a possible process to explain the high methyl-Hg proportions in the samples of the upper water layers but we agree with the reviewer that we did not show this directly and it is not our point. We discussed also other possible explanations for the relatively high seston MeHg concentration in our manuscript.

5. L235. But peak concentrations of MeHg in seston occur in the suboxic RTZ on 4 of the 5 dates when the lake was strongly stratified. On the remaining date, seston MeHg concentrations are highest in the upper hypolimnion. During stratification, MeHg is never highest in the oxic epilimnion. If anything, these finding suggest that MeHg production is associated with microbial respiratory pathways that are less energy efficient than O2 reduction (e.g. sulfate reduction, Fe reduction). Revise. R1 L264-267. The revised text is better ( more aligned with the data), but it now argues against their premise that methylation is occurring mainly in anoxic microniches within decaying seston as it settles. When the classic redox sequence has set up in the water column, the anaerobic microbes that possess the hgcAB genes can produce MeHg, but they don't have to reside within anoxic microniches in settling POM. Nothing in the data presented in this paper indicates or proves that they do. Instead, Mn, Fe and SO4 reducers may simply set up shop at the optimum depth and utilize the flow of nutrients and terminal electron acceptors from above (or below). This may occur in sediments or the water column. None of this is news, and none of it necessarily involves anoxic microniches in seston.

We cannot completely follow the reviewer here. The sentence in Line 235 of the original manuscript has been removed and the section has been clarified. It is not clear to which part the reviewer is referring his comment "..argues against their premise that methylation is occurring mainly in anoxic microniches within decaying seston as it settles". Our highest MeHg concentration in the seston is above the highest Mn concentration in the water column and far above the beginning of Fe reduction. Our data indicates that there is no sulfate reduction and thus no SO4 reducers within the water phase. Stratification omit mixing from the sediment and layers below the RTZ to layers above and into the RTZ. Thus, it is likely that there is another cause for the high MeHg concentration in the seston than Mn, Fe and SO4 reducers using terminal electron acceptors from the water phase.

Besides, to our knowledge there is only one paper that found the hgcA gene in manganese-reducing bacteria (Peterson et al., 2020), but their ability to actually methylate HgII still needs to be demonstrated.

6. L240-245. Alternatively, low MeHg during high productivity may reflect biodilution in the larger phytoplankton biomass (i.e. parental seston). Lacking sound data, one can't distinguish zooplankton bias from biodilution in microplankton, and neither necessarily point to sestonic microniches. Revise. I'm not convinced by the arguments in R1.

We don't know how to clarify this part based on this comment.

7. L255-263. They could also be explained by the presence of free-water microbes that possess the methylation gene pair hgcAB and occupy a region below the O/A boundary. DOM rather than POM could be their carbon source. Revise. R1 L308-324. This section remains highly speculative, and absent more rigorous investigation, alternative hypotheses can't be evaluated. The authors "assumption" that their explanation is the correct one isn't convincing

We agree with the reviewer that our hypotheses are speculative because we cannot completely prove them or exclude them with our data. But we discussed possible explanations aligned with our data. We conclude that the high MeHg concentrations at the RTZ could be explained by settling seston that aggregates within the RTZ. Settling particles such as this may form anoxic microniches, providing a thin vertical layer of high Hg methylation and biological activity, as suggested in other studies (Gascón Díez et al., 2016; Schartup et al., 2015; Ortiz et al., 2015; Gallorini and Loizeau, 2022).

8. L275-284. Sestonic MeHg in the 20% range is not atypical for unpolluted temperate lakes. What's unusual is the very low %MeHg in April. R1. L350-358. The mention of O2 fluxes into settling particles again assumes we are dealing with nonliving POM, but it's more likely that the "seston" collected in the plankton net comprises live organisms. The conflation of plankton with "lake snow" or dead POM aggregates is a conceptual problem throughout this paper

The reviewer is referring to L 315-320 in the revised manuscript. For the discussion of seston and living organisms in the particles see description above.

We propose, that the particles in the surface waters do not form aggregates (like in the RTZ) that are large enough to provide an ideal environment for mercury methylation in these high O₂ environment.

We will include the following paragraph in our manuscript:

*"The conditions in the RTZ in Lake Ölper are favourable to form anoxic micro-niches, whereas at the surface layer, oxygen concentrations related to the photosynthetic activity of the plankton are assumingly too high and the size of the particles too small for the formation of anoxic micro-niches"*

9. L346. Actually, this was first shown in Little Rock Lake, which is only 10m deep, and subsequently in many other lakes in this depth range. Not just deep oligotrophic lakes(but the eutrophic part may be right). R1. L346-347. This has not been addressed. The text remains unchanged and it seems like an attempt to oversell the novelty of this research.

We would like to include these references and lakes as they can improve our MS. Unfortunately, the reviewer missed to give us any reference. It is not clear for us to which paper the reviewer is referring to.

10. L346-end. Note that the range of Hg and MeHg in the seston of this eutrophic lake is on the low end of seston data reported for mesotrophic to oligotrophic North American lakes, both for MeHg concentration and %MeHg. High productivity is not necessarily conducive to abnormally high rates of MeHg accumulation in bioseston. In fact, most data suggest the opposite due to biodilution. It may be true that higher amounts of OM decomposition in eutrophic lakes does indeed exacerbate O2 depletion and enhance methylation in suboxic water, but that was not measured here. It seems that the most you can say with the data presented here is that the opposing forces of high biodilution and high decomposition need to be reconciled before addressing the impact of climate change. Revise R1 L349-357. This text also remains unchanged and it continues to promote the importance of anoxic microniches despite the fact that there is no direct evidence. The authors make claims about rates of methylation and demethylation without any rate determinations. Entirely unsupported speculations.

The main idea of our study is to show how THg and MeHg concentrations and proportions change during organic matter decay in sinking seston in the water phase during algae blooms and this approach is new, to our knowledge. We suggested the formation of MeHg in seston microniches as a possible process to explain the high methyl-Hg proportions in the samples of the upper water layers but we agree with the reviewer that we did not show this directly and it is not our point. We discussed other possible explanations for the seston MeHg concentration in our manuscript. However, our data suggest that free-water microbial Hg methylation is rather not the dominant process here as high MeHg concentration only occur during times of a pronounced RTZ (compare April when production is already high but MeHg is low because redox-zonation is not yet established).

We cannot follow the reviewer. We did not include any methylating and demethylating rate in our manuscript.

**References**

Alldredge, A. L. and Cohen, Y.: Can microscale chemical patches persist in the sea? Microelectrode study of marine snow, fecal pellets, Science (New York, N.Y.), 235, 689–691, doi:10.1126/science.235.4789.689, 1987.

Gallorini, A. and Loizeau, J.-L.: Lake snow as a mercury methylation micro-environment in the oxic water column of a deep peri-alpine lake, Chemosphere, 299, 134306, doi:10.1016/j.chemosphere.2022.134306, 2022.

Gascón Díez, E., Loizeau, J.-L., Cosio, C., Bouchet, S., Adatte, T., Amouroux, D., and Bravo, A. G.: Role of Settling Particles on Mercury Methylation in the Oxic Water Column of Freshwater Systems, Environ. Sci. Technol., 50, 11672–11679, doi:10.1021/acs.est.6b03260, 2016.

Lenz, J.: Seston and Its Main Components, in: Microbial Ecology of a Brackish Water Environment, Billings, W. D., Golley, F., Lange, O. L., Olson, J. S., Rheinheimer, G. (Eds.), Ecological Studies, Springer Berlin Heidelberg, Berlin, Heidelberg, 37–60, 1977.

Ortiz, V. L., Mason, R. P., and Ward, J. E.: An examination of the factors influencing mercury and methylmercury particulate distributions, methylation and demethylation rates in laboratory-generated marine snow, Mar. Chem., 177, 753–762, doi:10.1016/j.marchem.2015.07.006, 2015.

Peterson, B. D., McDaniel, E. A., Schmidt, A. G., Lepak, R. F., Janssen, S. E., Tran, P. Q., Marick, R. A., Ogorek, J. M., DeWild, J. F., Krabbenhoft, D. P., and McMahon, K. D.: Mercury Methylation Genes Identified across Diverse Anaerobic Microbial Guilds in a Eutrophic Sulfate-Enriched Lake, Environ. Sci. Technol., 54, 15840–15851, doi:10.1021/acs.est.0c05435, 2020.

Schartup, A. T., Balcom, P. H., Soerensen, A. L., Gosnell, K. J., Calder, R. S. D., Mason, R. P., and Sunderland, E. M.: Freshwater discharges drive high levels of methylmercury in Arctic marine biota, Proc. Natl. Acad. Sci. U.S.A, 112, 11789–11794, doi:10.1073/pnas.1505541112, 2015.

Shanks, A. L. and Reeder, M. L.: Reducing microzones and sulfide production in marine snow, Mar. Ecol. Prog. Ser., 96, 43–47, 1993.

Yigiterhan, O., Al-Ansari, E. M., Nelson, A., Abdel-Moati, M. A., Turner, J., Alsaadi, H. A., Paul, B., Al-Maslamani, I. A., Al-Ansi Al-Yafei, M. A., and Murray, J. W.: Trace element composition of size-fractionated suspended particulate matter samples from the Qatari Exclusive Economic Zone of the Arabian Gulf: the role of atmospheric dust, Biogeosciences, 17, 381–404, doi:10.5194/bg-17-381-2020, 2020.